# Symptoms of Prenatal Depression Associated with Shorter Telomeres in Female Placenta

**DOI:** 10.3390/ijms22147458

**Published:** 2021-07-12

**Authors:** Isabel Garcia-Martin, Richard J. A. Penketh, Samantha M. Garay, Rhiannon E. Jones, Julia W. Grimstead, Duncan M. Baird, Rosalind M. John

**Affiliations:** 1Division of Biomedicine, Cardiff School of Biosciences, Cardiff University, Cardiff, Wales CF10 3AX, UK; isabel.garciamartin@swansea.ac.uk (I.G.-M.); GaraySM@cardiff.ac.uk (S.M.G.); 2Department of Obstetrics and Gynaecology, University Hospital Wales, Cardiff, Wales CF14 4XW, UK; Richard.Penketh@wales.nhs.uk; 3Division of Cancer and Genetics, Cardiff School of Medicine, Cardiff University, Cardiff, Wales CF14 4XW, UK; JonesR47@cardiff.ac.uk (R.E.J.); SkinnerJW@cardiff.ac.uk (J.W.G.); BairdDM@cardiff.ac.uk (D.M.B.)

**Keywords:** telomere shortening, prenatal depression, placenta, sex differences

## Abstract

Background. Depression is a common mood disorder during pregnancy impacting one in every seven women. Children exposed to prenatal depression are more likely to be born at a low birth weight and develop chronic diseases later in life. A proposed hypothesis for this relationship between early exposure to adversity and poor outcomes is accelerated aging. Telomere length has been used as a biomarker of cellular aging. We used high-resolution telomere length analysis to examine the relationship between placental telomere length distributions and maternal mood symptoms in pregnancy. Methods. This study utilised samples from the longitudinal Grown in Wales (GiW) study. Women participating in this study were recruited at their presurgical appointment prior to a term elective caesarean section (ELCS). Women completed the Edinburgh Postnatal Depression Scale (EPDS) and trait subscale of the State-Trait Anxiety Inventory (STAI). Telomere length distributions were generated using single telomere length analysis (STELA) in 109 term placenta (37–42 weeks). Multiple linear regression was performed to examine the relationship between maternally reported symptoms of depression and anxiety at term and mean placental telomere length. Results: Prenatal depression symptoms were significantly negatively associated with XpYp telomere length in female placenta (B = −0.098, *p* = 0.026, 95% CI −0.184, −0.012). There was no association between maternal depression symptoms and telomere length in male placenta (B = 0.022, *p* = 0.586, 95% CI −0.059, 0.103). There was no association with anxiety symptoms and telomere length for either sex. Conclusion: Maternal prenatal depression is associated with sex-specific differences in term placental telomeres. Telomere shortening in female placenta may indicate accelerated placental aging.

## 1. Introduction

During pregnancy, women are highly vulnerable to major depression [1] and 10–15% experience at least one major depressive episode associated with the increased risk of morbidity to both mother and child [2]. Prenatal depression is co-morbid with anxiety, which can impact up to 25% of pregnancies, and both mood disorders are strongly linked to postpartum depression, all of which have a negative effect on child development in the short and longer term [3,4]. Every year in the UK > 100,000 babies are born exposed in utero to maternal depression, and considerably more are exposed to maternal anxiety. Importantly, recent surveys suggest that the rates of both depression and anxiety are increasing [5,6,7]. Maternal mood disorders represent a considerable clinical, financial and emotional burden to society [8]. A number of risk factors have been identified including previous history of poor mental health and socio-economic deprivation [9]. However, the biological mechanisms linking mood symptoms to adverse outcomes in children are unknown.

One biological factor linked to prenatal adversity is shortened telomere length. Telomeres are located at the ends of mammalian chromosomes and function to maintain genomic integrity [10,11]. Human telomeres consist of the hexameric DNA sequence, TTAGGG, tandemly repeated in arrays which vary in length depending on the tissue type and the age of the individual [10,12]. An individual’s telomere length is set at conception through the combined contribution of paternal and maternal telomere lengths inherited through the sperm and oocyte [13,14]. Whilst telomere length is maintained by the action of the enzyme telomerase in stem cell compartments [15,16], the majority of human somatic tissues do not express sufficient telomerase and thus exhibit erosion with ongoing cell division [12]. The gradual decline in telomere length over the individual’s lifespan is thought to play a role in the reduction of cellular viability via the induction of replicative senescence as a function of age [17]. Factors that accelerate or protect against telomere shortening during pregnancy may result in individuals being born whose telomere lengths differs from their chronological age [18]. Maternal age has been positively correlated with Telomere Restriction Fragment (TRF) length in newborns white blood cells [19]. In addition, placental telomere length decreases with gestational age and is influenced by parity, as multiparity appears to delay placental telomere shortening [20,21]. Maternal smoking, body mass index, stress, poor nutrition and disease status have been linked to shorter telomeres measured in cord blood DNA [22]. In adults, chronic psychosocial stress [23], major depression [24], chronic mood disorder [25] have been linked to shorter telomeres in white blood cells. Symptoms of depression reported in the first or second trimester of pregnancy have been linked to shorter cord blood telomeres in male infants but not female infants [26].

Few studies have examined telomere length in the placenta. The placenta is a fetally derived tissue that develops in concert with the fetus and is exposed to essentially the same environment [27], displaying low levels of telomerase activity and telomere shortening [28,29]. We recently reported shorter telomeres in placenta from pregnancies complicated by gestational diabetes, and this was the case only for male placenta. Here, we explored the relationship between telomere length profiles in placenta and maternally reported symptoms of depression and anxiety at term using measures of telomere length generated by high-resolution telomere length analysis (STELA). 

## 2. Methods

### 2.1. Cohort

The Grown in Wales (GiW) study [6] is a longitudinal birth cohort based in South-East Wales, United Kingdom that began in September 2015 and ended recruitment in November 2016. The basis of recruitment was attendance of a pre-surgical appointment prior to an elective caesarean section (ELCS) at the University Hospital of Wales during the specified time period with the stated criteria of women aged between 18 and 45 with a singleton term pregnancy without fetal abnormalities or infectious diseases. ELCS was chosen as the mode of delivery to facilitate the collection of high-quality biological samples which included maternal serum and saliva at recruitment as well as placenta and cord blood on delivery. The major indication for ELCS was a previous caesarean. We previously reported no significant difference in late antenatal mood scores between participants with varying indications for ELCS [6]. All participants were recruited by two trained research midwives. Women were not clinically assessed for depression, anxiety or stress. A total of 355 women were originally recruited into the study, seven of whom later withdrew. Full ethical approval for the GiW study was obtained from the Wales Research Ethics Committee REC2 reference 15/WA/0004. Research was carried out in accordance with the principles of the Declaration of Helsinki as revised in 2008. Written informed consent was obtained from all the participants at recruitment.

### 2.2. Materials

#### 2.2.1. Maternal Demographics and Birth Outcomes

Maternal lifestyle and demographics were reported by the mother in the questionnaire at recruitment which was 1–4 days prior to ELCS. Data included ethnicity, education, income, age, body mass index (BMI), pregnancy complications and whether they smoked or drank alcohol during their pregnancy. Welsh Index of Multiple Deprivation (WIMD) 2014 scores were calculated from anonymised postcodes [30]. Delivery information, fetal and placental biometry, and body mass index at initial booking was recorded from medical notes. Gestational age at the prenatal assessment was the same as gestational age at ELCS. Prevalence of obstetric risk factors are provided in Appendix A. 

#### 2.2.2. Maternal Depression and Anxiety Symptoms

Participants completed two self-reporting mood questionnaires at recruitment 1–4 days before their ELCS. Depression was measured using the Edinburgh Postnatal Depression Scale (EPDS) which comprises of ten questions each scored between zero and three, with total scores 13 and above indicating probable depression [31]. Anxiety was measured using the Trait subscale of the State Trait Anxiety Inventory (STAI) to measure general anxiety levels [32,33]. This subscale contains 20 questions scored between one and four, with final scores of 40 and above indicating high anxiety levels [32]. Missing data was addressed using participant level mean substitution for those missing <20% of data. 

#### 2.2.3. Placental Biopsies

Biopsies were collected from term placenta (37–42 weeks) by trained research midwives within two hours of delivery. For each placenta, 3–5 chorionic villous samples were taken 1 cm below the surface from the maternal side of the placenta at sites midway between the cord insertion and the lateral edge. Samples were washed three times in phosphate buffered saline to remove maternal blood and stored in RNAlater at −80 °C. 

#### 2.2.4. STELA

Placental telomere length distributions were obtained using the STELA assay as described previously [34]. Briefly, genomic DNA was extracted from approximately 40 mg of placental tissue consisting of a combined tissue sample from 3–5 separate biopsies by a standard proteinase K and phenol/chloroform protocol [35] with all samples processed in the same way by a single researcher to minimise technical variability. Diluted genomic DNA samples were PCR amplified using telorette2, teltail and the XpYp telomere specific primer XpYpE2. Amplified telomeric DNA fragments were resolved on 0.5% agarose gels and detected by Southern hybridisation using a TTAGGG repeat probe α-^33^P dCTP labelled (Perkin Elmer) before visualisation using a Typhoon FLA 9500 phosphoimager (GE Healthcare Life Sciences, United Kingdom). Telomere length distributions were determined using the ImageQuant software (GE Healthcare Life Sciences, United Kingdom). Telomere measures were generated for N = 133/337 of the placental biopsies collected. For this study, we selected 30 samples with EPDS scores ≥ 13 as our “depressed” group and 79 with EPDS scores < 13 as our non-depressed “low mood score” group excluding samples with a diagnosis of preeclampsia or gestational diabetes, those with gestational age < 37 weeks and those not ultimately delivering by ELCS. Two of the 109 participants did not have a STAI or an EPDS value.

### 2.3. Statistical Analysis

#### 2.3.1. Research Question

Is telomere length affected by prenatal depression? Is there a sex-specific effect of prenatal depression on telomere length?

#### 2.3.2. Hypothesis

Placentas from women exposed to prenatal depression will exhibit shorter telomeres than placentas from control women.

#### 2.3.3. Statistical Steps

All statistical analysis was performed using SPSS 26.0 for Macintosh and GraphPad Prism 9. Data are expressed as means with standard deviation, or as numbers (%). Normality was assessed using Shapiro–Wilk test, histograms and normal Q–Q plots (Figure 1). Mean placental telomere length was normally distributed (*p* > 0.05) as determined by Shapiro–Wilk test. Differences between low mood score and high mood score groups (EPDS < 13 and EPDS ≥ 13) were assessed using χ2 test or Student’s *t*-test. Relationships between the main dependent variable (mean telomere length) and other confounding variables were analysed by Pearson or Spearman correlation. To assess the relationship between mean telomere length and maternal mental health scores (EPDS and STAI), unadjusted and adjusted linear regression was performed. To test the main effect of maternal perinatal mental health on telomere length, unadjusted linear regression was performed. This included as the dependent variable mean telomere length, and EPDS or STAI as the independent variable. 

A list of relevant covariates controlled for in the linear regression included maternal age, gestational age, parity, BMI, smoking during pregnancy, alcohol during pregnancy, WIMD score and EPDS or STAI. These variables were chosen based on the established literature in the introduction and methods section. All unadjusted and adjusted linear regressions were run separately for participants who had girls or boys.

## 3. Results

### 3.1. Association between Telomere Length and Potential Confounders

As in our previous studies [34,36], this study applied STELA to measure the lengths of the XpYp telomeres located at the end of the pseudoautosomal region. Placental telomere length distributions were analysed for term placental samples obtained from 109 Grown in Wales participants delivering by ELCS excluding the pregnancy complications of gestational diabetes and preeclampsia. There was no significant relationship (*p* > 0.05) between mean telomere and the obstetric risk factors listed in Appendix A. The association between potential confounders (Table 1) including maternal age, gestational age, parity, smoking, alcohol consumption, BMI and WIMD score with telomere length was also tested. Average telomere length was correlated with WIMD score with a Spearman’s rho coefficient of 0.216, *p* < 0.05. WIMD score ranks all small areas in Wales from 1 (most deprived) to 1909 (least deprived) and is therefore an indicator for socio-economic status.

### 3.2. Analysis of High and Low Mood Score Groups

In our previous study on GDM, we detected significantly shorter telomeres and significantly more telomeres under 5 kb in male placenta with a group size of 38 controls versus 10 GDM [36]. Using data from this study, we calculated we would need a sample size of 13 in each group (α = 0.05 and a power of 0.8). Of the 109 sample measures analysed in this study *n* = 55 were male and *n* = 54 were female placentas. Of these, 30 were from pregnancies where women reported significant depression symptoms just prior to the ELCS with an EPDS score of ≥13 (high mood score), with 17 samples from male placenta and 13 samples from female placenta. Of these 30 samples, 25 women also scored above the cut-off for STAI with a score of ≥40, i.e., for most of these mothers there was evidence of both depression and anxiety consistent with the significant association between EPDS and STAI with a Spearman’s rho coefficient of 0.877**, *p* < 0.01. The remaining 79 samples, with 38 samples from male placenta and 41 samples from female placenta, formed a non-depressed “low mood score” group. There were no differences for any of the listed characteristics between the two groups (*p* > 0.05; Table 2).

### 3.3. Multiple Linear Regression

In order to investigate the association between mean telomere length and perinatal anxiety and depression, multiple linear regression was performed. There were no significant associations when data from all participants (*n* = 109) was analysed unadjusted (Table 3). Disparate sex-specific features of prenatal vulnerability have been reported in relation to specific pregnancy conditions [26,36,37,38,39]. Data from male and female samples were therefore analysed separately. The analysis was then adjusted for maternal age, gestational age, parity, BMI, smoking during pregnancy, alcohol during pregnancy and WIMD covariates (Figure 2 and Table 3). There was no statistical difference in mean placental telomere length between male and female placenta (mean = 7.97, SD = 1.66 versus mean = 8.13, SD = 1.47; *p* = 0.60). EPDS scores were significantly negatively associated with shorter mean placental telomere length for female infants only (*p* = 0.026; Figure 3A). 35% of the variance in female placental telomere length was accounted for by the EPDS score and the abovenamed covariates.

## 4. Discussion

Our study uncovered an association between maternally reported symptoms of depression recorded via an EPDS questionnaire completed just prior to delivery and placental telomere length. EDPS scores were significantly negatively associated with mean placental telomere length in female placenta only. This relationship remained after adjusting for factors known or suspected to influence telomere length including maternal age, gestational age, parity, smoking, alcohol, BMI and WIMD score. In contrast, STAI scores (anxiety) were not associated with placental telomere length.

The placenta is derived from the same genome as the fetus and exposed to the same factors but, due to its position between the maternal and fetal circulation, may be more impacted by adversity in pregnancy. We previously reported that term placenta show remarkable heterogeneity in placental telomere lengths [34]. We also reported that the length and distributions of telomeres were not influenced by sampling site, mode of delivery or fetal sex [34]. Using the same STELA approach, we more recently reported shorter telomeres in male placenta from pregnancies where mothers were diagnosed with gestational diabetes, but not medically treated with metformin or insulin [36]. Shorter placental telomeres have previously been associated with being born small for gestational age or growth restricted [40,41,42,43] and preeclampsia [44]. Higher maternal reporting of adverse childhood experiences has also been associated with shorter placental telomeres [45]. These data all lend support to the idea that pregnancy complications impact the maintenance of telomere integrity in the placenta which may have relevance for placental function and fetal wellbeing. 

The major finding of our study was the negative association between prenatal EPDS score and telomere length in female placenta. Higher maternally reported symptoms of depression, but not anxiety, were associated with significantly shorter telomeres. In contrast, there was no association between EDPS score and telomere length in male placenta. The absence of association in male placenta was not due to a lack of statistical power as analysing all the placenta combined did not reveal an association despite twice the number of samples in the analysis (*n* = 109 all samples v *n* = 54 female samples v *n* = 55 male samples). EPDS scores are used to indicate symptoms of depression with scores of 13 and above predicting an episode of clinical depression based on diagnostic criteria in the postpartum period [31,46,47]. Maternal stress during pregnancy has been associated with shorter cord blood telomeres [48,49,50] with one retrospective study suggesting a greater impact on females [51]. Air pollution is another adversity linked to shorter cord blood telomeres with a stronger association in girls [52]. However, in a recent study of 151 newborns, female infant cord blood telomere length was not associated with any factor tested. This study reported associations between shorter male cord blood telomeres and maternal smoking, higher body mass index, maternal sexual abuse in childhood and elevated depressive symptoms in pregnancy. Longer cord blood telomere length was associated with higher maternal educational attainment and household income in pregnancy, and greater maternal familial emotional support in childhood [26]. There are a number of differences between this study and ours, the most obvious being the nature of the samples measured (placenta versus cord blood), the technical approach (STELA versus real-time polymerase chain reaction) and the population (Welsh White versus North American White, Hispanic and Black/Haitian). In addition, Bosquet Enlow et al. measured depressive symptoms by EPDS questionnaire during the first or second trimester [26] whereas our participants completed this questionnaire just prior to term. Bosquet Enlow et al. also excluded participants endorsing drinking more than seven alcoholic drinks per week. These differences may account for our different findings. An interesting alternative possibility is that telomere length in fetus may be preserved in females exposed to stressors in pregnancy at the cost of placental telomeres while the reverse may be true in males. Recently in a new study on 146 newborns [53] reported no association between annual household income and telomere length. This study also reported a positive association between newborn relative telomere length and pregnancy-related anxiety symptoms (PRAS), depressive symptoms during pregnancy (EPDS), general anxiety symptoms (STAI), and self-reported depression prior to the current pregnancy. These findings suggested a positive and adaptive effect of maternal stress on fetal telomere biology among males, which contradicts their previous reported findings that correlated elevated depressive symptoms in pregnancy and shorter cord blood telomeres in boys. This variability in similar studies emphasises the need to dig into the influence of pre-conception factors on fetal telomere biology, as well as to define the clinical significance of shorter and longer telomere length at birth. Moreover, the simultaneity of certain factors and maternal physical and mental health issues may rescue the detrimental effect of the latter on fetal telomere biology. It can be hypothesised that compensatory factors, such as socioeconomic status, may ameliorate the negative impact of maternal depressive and anxiety symptoms on fetal telomere length. 

In our study we did not find an association between shorter placental telomeres and depressive symptoms in male placenta, but we did observe an association between higher WIMD score and longer telomeres. WIMD scores are generated using postcode information and reflect the Welsh Government’s official measure of relative deprivation within small areas in Wales. A number of measures feed into WIMD including income, employment, health, education, access to services, housing, community safety and physical environment. While non-deprived people can live in deprived areas, and deprived people can live in non-deprived areas, higher WIMD scores are generally indicative of lower overall levels of deprivation. Our data is consistent with the previous findings that higher maternal education and income provide protection against telomere shortening in males [26]. Moreover, a recent study by Martens et al. [54] found that higher parental socioeconomic status was associated with longer cord blood telomere length and placental telomere length in boys, but not in girls. These findings indicate that social economic factors likely play an important role in influencing cellular longevity in the exposed individuals.

### Limitations

One limitation of our study is that we are relying on mood scores from self-reported questionnaires recorded 1–4 days before an ELCS. For our full GiW study cohort, we previously reported a highly significant association between EPDS/STAI scores and mental health history, and also between term EPDS/STAI scores and those obtained at three points after delivery for a subset of participants [6] which is reassuring. However, although self-reported questionnaires have a high acceptance by women who are thought to feel less constrained in responding as stated by the WHO (2008), a full clinical evaluation would provide further validation of our findings. Some of the factors controlled for were also self-reported including alcohol consumption, although we found that reporting for these factors was highly similar between the questionnaires and the medical notes [6]. Another limitation of this study was the nature of the cohort which had a restricted participant demographic. While minimising heterogeneity within a cohort study has value in uncovering subtle relationships with smaller cohort numbers, our findings may not be generalizable to other populations. A further limitation is that we do not have the parental telomere lengths. It is possible that our findings are an indicator of telomere length at conception and not exposures during pregnancy, although if this were the case it would be more challenging to interpret the sex-specific findings. A final limitation is that our analysis was restricted to the XpYp telomeres located at the end of the pseudoautosomal region. While this region segregates independently of sex with similar telomere distributions to autosomes [55], it is possible that our findings are confined to the telomeres we analysed. Further work is required to validate our findings in other cohorts with a greater diversity of participants and extend this analysis to other chromosomes. It will also be important to compare placental and cord blood telomere lengths from matched samples to establish the potential for inheritance in other fetally derived tissues.

## 5. Conclusions

In summary, we discovered that mothers reporting higher symptoms of depression near term gave birth to daughters with relatively shorter placental telomeres. We did not observe a similar relationship between mood symptoms and placental telomeres when mothers gave birth to sons. However, we did observe longer male placental telomeres in mothers residing in more affluent areas. These findings contribute to our understanding of the sex-specific outcomes observed for children exposed in utero to maternal depression.

## Figures and Tables

**Figure 1 ijms-22-07458-f001:**
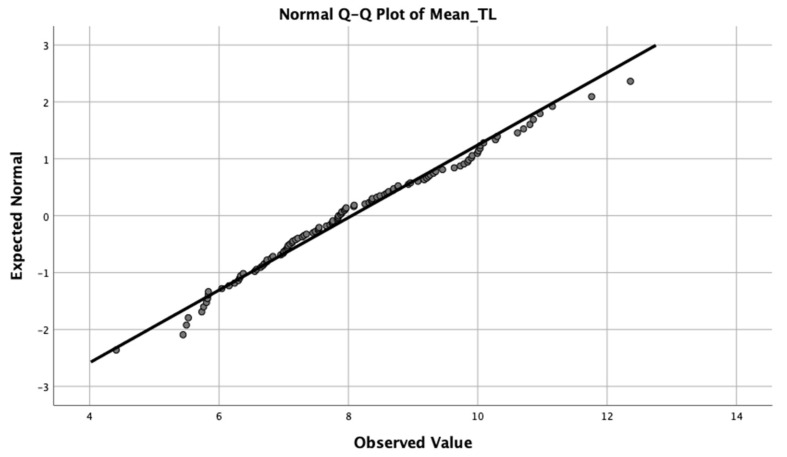
Q–Q plot of mean telomere length.

**Figure 2 ijms-22-07458-f002:**
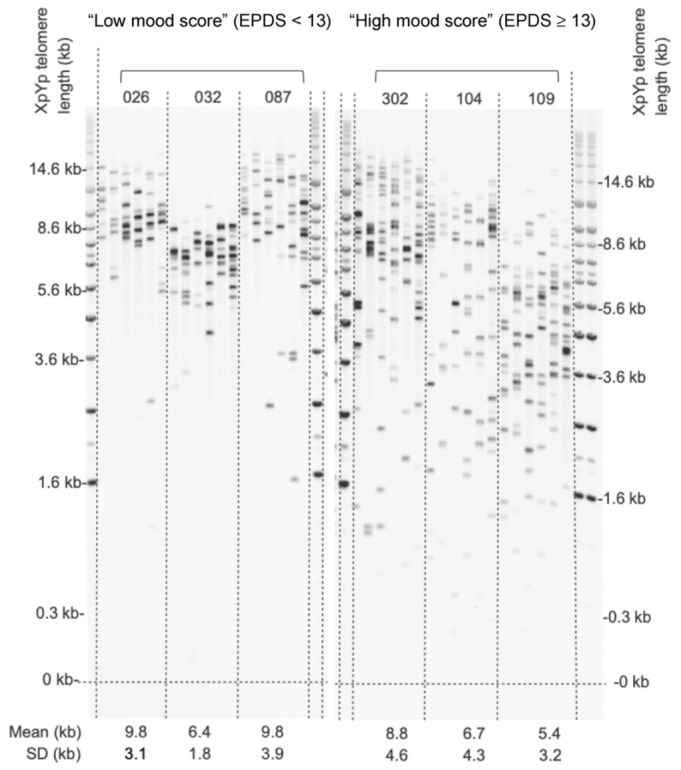
Representative STELA. Images of two autoradiographs are shown alongside molecular weight markers. The left panel shows STELA PCR reactions from three female placenta samples from the “low mood score” (EPDS < 13) group with 6 reactions run for each sample. The right panel shows comparable data for “high mood score” (EPDS ≥ 13) group. Mean telomere lengths ± standard deviation (SD) are given below the lanes for each sample. The coefficient of variation of all samples was <1 indicating low variance.

**Figure 3 ijms-22-07458-f003:**
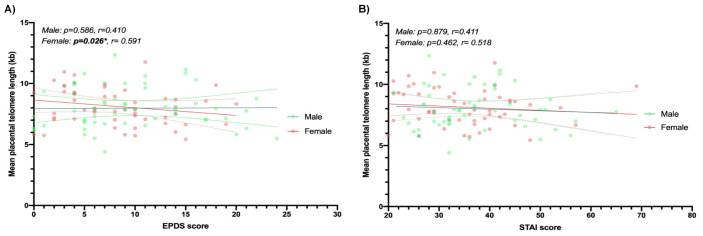
Telomere length differences in placenta according to mood scores (EPDS and STAI) for males (*n* = 55) and females (*n* = 54). The association between maternally reported depression symptoms (EPDS score) and mean placental telomere length (**A**), and the association between maternally reported anxiety symptoms (STAI score) and mean placental telomere length (**B**) are shown as linear regressions.

**Table 1 ijms-22-07458-t001:** Association between telomere characteristics and potential confounders. Number (%) is shown; *p*-values were assessed with Pearson or Spearman rho correlation; *n* = 109 samples.

Potential Confounder	Association with Mean Telomere Length
Maternal age	r = 0.09 CI = −0.030, 0.078 *p* = 0.37
Gestation age	r = 0.15 CI = −0.100, 0.832 *p* = 0.12
Parity	r = 0.09 CI = −0.173, 0.479 *p* = 0.35
Smoking 14/109 (12.9%)	r = −0.00 CI = −0.765, 1.023 *p* = 0.776
Alcohol 8/109 (7.3%)	r = 0.11 CI = −0.421, 1.85 *p* = 0.214
BMI	r = −0.03 CI = −0.056, 0.093 *p* = 0.792
WIMD	r = 0.22 CI = 0.000, 0.001 ***p* = 0.027**

BMI = body mass index; WIMD = Welsh Index of multiple deprivation.

**Table 2 ijms-22-07458-t002:** Key characteristics of the study participants. Mean (SD)/range or number (%) is shown; *p*-values were assessed using independent samples *t*-test or χ^2^ test. Due to missing values some numbers do not add up to 100%.

Characteristics	Low Mood Score Group: EPDS < 13 Group (*n* = 79)	High Mood Score Group: EPDS ≥ 13 Group (*n* = 30)	*p*-Value
Caucasian ethnicity	73 (92%)	29 (96%)	0.52
Parity:			0.82
Primiparous	14 (17.7%)	7(23.3%)	
Multiparous	65 (82.2%)	23 (76.7%)	
Maternal age	32 (5.51)/19–44	30 (5.68)/20–39	0.31
ELCS	79 (100%)	30 (100%)	N/A
Birth weight (g)	3491 (620)/2260–5080	3488 (499)/2460–5110	0.98
Gestational age (weeks)	39 (0.61)/38–41	38 (0.68)/37–41	0.12
Placental weight (g)	663 (129)/376–941	671 (150)/455–1060	0.76
Fetal sex:			0.42
Female	38 (48%)	17 (56%)	
Male	41 (52%)	13 (44%)	
Smoking during pregnancy	9 (11.4%)	5 (16.7%)	0.46
Alcohol during pregnancy	5 (6.3%)	3 (10%)	0.51

ELCS = elective caesarean delivery.

**Table 3 ijms-22-07458-t003:** Analysis of the association between mean telomere length and maternal mood scores using unadjusted and adjusted multiple linear regression models. Adjusting for maternal age, gestational age, parity, BMI, smoking during pregnancy, alcohol during pregnancy, WIMD score and EPDS/STAI. A *p*-value < 0.05 is considered statistically significant.

Mood Scores	All			Male			Female		
	B	95%CI	*p*	B	95%CI	*p*	B	95%CI	*p*
Unadjusted linear regressions									
EPDS	−0.025	−0.079, 0.028	0.346	0.002	−0.073, 0.078	0.949	−0.063	−0.143, 0.017	0.121
STAI	−0.016	−0.046, 0.015	0.305	−0.013	−0.060, 0.034	0.582	−0.017	−0.059, 0.024	0.399
Adjusted linear regressions									
EPDS	−0.004	−0.061, 0.052	0.884	0.022	−0.059, 0.103	0.586	−0.098	−0.184, −0.012	**0.026**
STAI	−0.002	−0.033, 0.030	0.914	−0.004	−0.054, 0.046	0.879	−0.016	−0.058, 0.027	0.462

EPDS, Edinburgh Postnatal Depression Scale; STAI, Spielberger State-Trait Anxiety Inventory; WIMD, Welsh Index of Multiple Deprivation; BMI, body mass index.

## Data Availability

For this specific study, there is a danger that sharing participant data might reveal participant identity due to the nature of the data, the specific recruitment dates and unique route of recruitment. The datasets used and analysed during the current study will therefore not be made publically available but will be available from the corresponding author on reasonable request.

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
