# Peer review of "Symptoms of Prenatal Depression Associated with Shorter Telomeres in Female Placenta"

_ijms, 2021, doi:10.3390/ijms22147458_

Round 1

Reviewer 1 Report

The paper by Garcia-Martin et al investigates the association between placental telomere lengths and the symptoms of maternal prenatal depression. The authors describe the association between prenatal depression and shorter telomere length in placentas of female new-borns. The study is interesting and the methods used are well described. I only have a few minor comments.

  1. Why were only women undergoing ELCS included? What were the indications for Caesarean sections?
  2. Why were only 109 placental samples available out of 355 included women to the study? What were the selection criteria? This is not clear from the manuscript. Why are there only 107 samples included in the Supplementary table 1.

Author Response

Thank you for the swift review of our submission for this special issue. Please find our responses and amendment to the manuscript as follows:

1. Why were only women undergoing ELCS included? What were the indications for Caesarean sections?

Response: In a previous pilot study where we recruited women earlier in pregnancy prior to mode of delivery being known, the collection of placental samples of sufficient quality for RNA analysis was found to be highly variable from natural labour. Moreover, we found that gene expression varied with mode of delivery (Janssen 2015). Consequently the Grown in Wales study focused on recruiting women undergoing ELCS to maximise the collection of placental and other biological samples of high quality. Indication for ELCS were given in Jansen 2018 and principally are due to previous caesarean delivery. Importantly, indication for ELCS was not associated with mood scores derived from questionnaires.  

Line 98: ELCS was chosen as the mode of delivery to facilitate the collection of biological samples which included maternal serum and saliva at recruitment as well as placenta and cord blood on delivery. The major indication for ELCS was a previous caesarean and we previously reported no significant difference in late antenatal mood scores between participants with varying indications for ELCS (Janssen 2018).

2. Why were only 109 placental samples available out of 355 included women to the study? What were the selection criteria? This is not clear from the manuscript.

Response: Apologies for not making this clear in our material and methods. We have a total of 133 placental telomere measures for the Grown in Wales study cohort, which is part of a larger study investigating placental telomere characteristics over two pregnancy cohorts. For the current study, we selected 30 samples with EPDS scores 13 and above, and 79 with EPDS <13 after excluding samples with a diagnosis of preeclampsia or gestational diabetes, those with gestational age <37 weeks and those not ultimately delivering by ELCS.

Line 146: Telomere measures were generated for N=133/337 of the placental biopsies collected. For this study, we selected 30 samples with EPDS scores ³13 as our “depressed” group and 79 with EPDS scores <13 as our non-depressed “low mood score” group after excluding samples with a diagnosis of preeclampsia or gestational diabetes, those with gestational age <37 weeks and those not ultimately delivering by ELCS. 

Line 191: analysed for term placental samples obtained from 109 participants delivering by ELCS excluding the pregnancy complications of gestational diabetes and preeclampsia.

3. Why are there only 107 samples included in the Supplementary table 1.

Response: Information for all the variables was only available for n=107 women as there was missing information for 2 participants regarding FGR diagnosis.

Line 524: Information for all the stated variables only available for n=107/109 participants.

Reviewer 2 Report

Manuscript ID: ijms-1264969

Title: " Symptoms of prenatal depression associated with shorter telomeres in female placenta”.

 Authors : Martin  et al .

Manuscript Type: Article

The manuscript of Martin   and colleagues reports a study aimed to investigate the  relationship between mood symptoms and placental telomeres. The authors have observed that maternal prenatal depression is associated with sex- specific differences in term placenta telomeres but however some specific criticisms should be addressed.

  1. In the “Methods” section (pag 2 line 92) it is important to specify if  355 women had diagnosis of  depression, anxiety or stress .
  2. I suggest to add in table 2 maternal lifestyle such as smoked or drank alcohol during their pregnancy
  3. In the “Results ” section (pag 5 line 189) it is important to specify the number of male placenta and female placenta of the 109 samples.
  4. In the “Results ” section (pag 5 line 196) the remaining 79 were all female placentas? It may be useful to add this information.
  5. Age-dependent shortening of telomere length has been shown in several studies. Interestingly, if the mean age-dependent shortening rate was very similar among low mood score” and  high mood score group. The authors could better discuss this in the discussion section.

Author Response

Thank you for the swift review of our submission for this special issue. Please find below our response to your comments and the amended text of our manuscript:

1. In the “Methods” section (pag 2 line 92) it is important to specify if 355 women had diagnosis of  depression, anxiety or stress .

Response: We have clarified that women were not clinically assessed for depression, anxiety or stress. We also mentioned this as a limitation in our discussion.

Line 103: The basis of recruitment was attendance of the clinic during the specified time period with the stated exclusions. Women were not clinically assessed for depression, anxiety or stress.

2. I suggest to add in table 2 maternal lifestyle such as smoked or drank alcohol during their pregnancy

Response: This information has been added and highlighted in yellow text

3. In the “Results ” section (pag 5 line 189) it is important to specify the number of male placenta and female placenta of the 109 samples.

Response: This information has been added and highlighted in yellow text

4. In the “Results ” section (pag 5 line 196) the remaining 79 were all female placentas? It may be useful to add this information.

Response: Apologies for the confusion – we have clarified the sex of the placenta and highlighted in yellow text

5. Age-dependent shortening of telomere length has been shown in several studies. Interestingly, if the mean age-dependent shortening rate was very similar among low mood score” and  high mood score group. The authors could better discuss this in the discussion section.

Response: We agree that this is a really interesting point that needs to be explored especially as some studies have reported age-associated differences in placental telomere characteristics while others have not. We did not find an association between either maternal age and telomere length, or gestational age and telomere length in our study but this is not surprising as our placenta were all from term pregnancies with a relatively limited range in maternal age. Moreover, for multifactorial analysis a considerably larger number of placental measures will be required. However, we hope that studies such as ours will contribute to much larger studies across multiple cohorts as STELA is a technique where the measures can easily be combined for the type of analysis mentioned.